# In-Silico Molecular Modeling Studies to Identify Novel Potential Inhibitors of HPV E6 Protein

**DOI:** 10.3390/vaccines10091452

**Published:** 2022-09-02

**Authors:** Moujane Soumia, Halima Hajji, Mohamed El Mzibri, Filali Zegzouti Younes, Bouachrine Mohammed, Benlyas Mohamed, Moualij Benaissa

**Affiliations:** 1Biochemistry of Natural Substances, Faculty of Science and Techniques, Moulay Ismail University, Errachdia 50003, Morocco; 2Molecular Chemistry and Natural Substances Laboratory, Faculty of Science, Moulay Ismail University, Meknes 52202, Morocco; 3EST Khenifra, Sultan Moulay Sliman University, Khenifra 23000, Morocco; 4Biology and Medical Research Unit, National Centre for Energy, Nuclear Sciences and Techniques (CNESTEN), Rabat 10001, Morocco; 5LABASE Laboratory, Faculty of Science of Meknes, Moulay Ismail University, Meknes 52202, Morocco

**Keywords:** cervical cancer, MD simulations, docking, virtual screening, E6-E6AP, ADMET properties

## Abstract

The etiological agent of some anogenital tract cancers is infection with the high-risk human papillomavirus (HPV). Currently, prophylactic vaccines against HPV have been validated, but the presence of drug treatment directed against the infection and its oncogenic effects remain essential. Among the best drug targets, viral oncoprotein E6 has been identified as a key factor in cell immortalization and tumor progression in HPV-positive cells. E6, through interaction with the cellular ubiquitin ligase E6AP, can promote the degradation of p53, a tumor suppressor protein. Therefore, suppression of the creation of the E6-E6AP complex is one of the essential strategies to inhibit the survival and proliferation of infected cells. In the present study, we proposed an in-silico approach for the discovery of small molecules with inhibitory activity on the E6-E6AP interaction. The first three compounds (F0679-0355, F33774-0275, and F3345-0326) were selected on the basis of virtual screening and prediction of the molecules’ ADMET properties and docking with E6 protein, these molecules were selected for further study by investigating their stability in the E6 complex and their inhibitory effect on the E6-E6AP interaction by molecular dynamics (MD) simulation. The identified molecules thus represent a good starting point for the development of anti-HPV drugs.

## 1. Introduction

Cervical cancer is the second most common cancer in women. More than “500,000” new cases are registered annually with an overall rate of 18.8 per 100,000 women. In 2020, it was an estimated 604,000 new cases and 342,000 deaths worldwide [1]. Currently, it is widely accepted that Human papillomavirus is the etiological agent of cervical cancer [2] and that high-risk (HR) HPVs are responsible for the occurrence and development of approximately 5% of all cancers and is associated with 30% of all pathogen-related cancers [3]. Human papillomaviruses (HPV) belong to the Papillomaviridae family. There are more than 200 distinct HPV genotypes known for their ability to infect mucous membranes and human skin epithelial cells HPVs are the etiological agent of the majority of the most common sexually transmitted viral infections in men and women and recent studies have shown that HPV can also affect fertility, pregnancy rates, and other health outcomes [4,5,6]. The viral genome consists of seven genes classified as early genes (E1, E2, E3, E4, E5, E6, and E7), which regulate viral transcription and genome replication, and two Late genes (L1 and L2) encode the structural proteins involved in capsid formation [7]. The HR-HPV16 is the causative agent of most cervical cancer cases. Its high oncogenic potential is mainly due to the oncoproteins E6 and E7 [8,9]. Scientific evidence has shown that p53 is the main target of the E6 oncoprotein. However, several studies have reported that E6 binds to a diverse range of cellular proteins playing a key role in the cell cycle control and regulation [10,11]. The E6 contains two zinc-binding domains with a conserved fold that are connected to each other by a helical linker The E6 amino-terminal zinc-binding domain and the carboxy-terminal zinc-binding domain have a globally conserved fold in the crystal [12]. The two zinc domains, together with an alpha helix tube connecting them, form a deep pocket in which the LXXLL peptide makes close contact [13]. During HPV-induced carcinogenesis, HR E6 oncoproteins induce p53 degradation. During E6-mediated p53 degradation, HR-HPV E6 proteins interact with the LxxLL motif of E6AP, LxxLL a 20-amino acid peptide in E6AP, resulting in the recruitment and polyubiquitination of p53. The LxxLL peptide isolated from E6AP is sufficient to render E6 susceptible to interaction with p53. [14]. the central pocket of E6 binds directly to the LxxLL motif of the ubiquitin ligase E6AP [15,16], which induce a conformational change in HR E6 proteins allowing the formation of a complex with p53. In this ternary complex, called E6/E6AP/p53, p53 is polyubiquitinated by E6AP and then degraded by a proteasome, thereby directly inhibiting apoptosis [17]. Of particular interest, HR E6 oncoproteins were reported to be involved in regulation of p53 gene transactivation and are able to abolish p53 transcriptional transactivation activity [18]. HR E6 oncoproteins can also interact with p300/CBP co-activators to control p53-dependent gene regulation [19].

Currently, three prophylactic vaccines have been approved and used effectively to prevent persistent viral infections and HPV-associated cervical lesions: Cervarix^®^2, which is effective for HPV genotypes 16 and 18 [20]; Gardasil^®^4, effective for HPV [21] genotypes 6, 11, and 16; and Gardasil^®^9 is effective for HPV genotypes 6, 11, 16, 18, 31, 33, 45, 52, and 58 [22]. Nevertheless, the vaccination program is not well established and, except in some developed countries, the vaccination is not provided in most countries. Thus, management of cervical cancer and precancerous lesions is still limited to the use of chemotherapeutic agents and/or the implementation of surgical and ablative techniques to remove the developed tumors. Both of these treatments are invasive, non-specific, and tend to be expensive, making their accessibility limited to millions of patients, especially in developing countries [23]. There is, therefore, an urgent need to develop accessible drug-based therapies targeting the onco-virus for specific treatment of HPV-associated diseases and better management of cervical cancer and precancerous lesions.

Thus, the present study was planned to identify small molecules from 6346 chemicals available in the Life Chemicals database (http://www.lifechemicals.com Format SDF accessed on 11 February 2020) with potential inhibitory activity against HPV E6. In this study, an integrated bioinformatics approach was used to identify lead compounds that could serve as inhibitors for the HPV E6 oncoproteins.

We identified three drug-like compounds, resulting in the discovery of several chemical entities that offer novel scaffolds that could be used as the core of new families of E6 HPV16 inhibitors. All three compounds were used in the 50 ns MDS study. Based on various parameters such as RMSD, and RMSF, we report (F0679-0355, F33774-0275, and F3345-0326) that they have the same backbone as lead compounds, which could serve as E6 HPV16 inhibitors. However, further in vitro and/or in vivo research is needed to validate the in-silico results

## 2. Material and Methods

### 2.1. Collection and Curation of the Chemical Library

Chemical structures of 6340 compounds with potential anti-tumor activity, targeting various types of cancer, such as prostate, breast cancers, leukemia, lymphoma, carcinoma, etc., were downloaded as 3D sdf files from the Life Chemicals Anti-Cancer Screening Library (https://lifechemicals.com Format SDF accessed on 11 February 2022) and prepared with Open babel [24].Lipinski’s rule, with 300 < Mw < 700 g/mol and 5 < Number of rotatable bonds < 12), was applied to filter out non-drug-like molecules [25].

### 2.2. Structure-Based Virtual Screening

Drug-like molecules were considered as a database to identify new compounds with high binding affinity and chemical complementarity to the E6/E6AP/p53 tetramer active site, using a virtual screen. The 3D structures of E6/E6AP/p53 (PDB code: 4xR8) were downloaded from the Protein Data Bank [26]. In order to prepare the selected protein for molecular docking, water and co-crystallized small molecules were removed. The protein contains eight chains; seven chains were removed, and the H-chain was particularly retained to make the calculations shorter and simpler, the non-polar hydrogens were added using discovery studio [27]. A grid box with a size of (x = 40; y = 80; z = 60) and a center of (x = 16.0; y = −32.0; z = −19.0) was defined to cover the ligand-binding site at 4xr8.

### 2.3. Molecular Docking

Selected molecules from the virtual screening were molecularly docked to the PDB target (4xr8). The Autodockvina [28], and MGL Tools [29] programs were run with their default settings.

### 2.4. Post Docking Analysis

Structures built on molecular docking results (E6/E6AP/p53) were examined separately. The molecules were prioritized by taking into account the binding energy of their highest-scoring conformation and then the top 10 candidates were visually inspected for their interactions with the active site. Their interactions with the active site residues were visually examined. Finally, the selected molecules were analyzed in more detail (ADMET prediction and molecular dynamics). Visual inspection of docking poses and analysis of protein–ligand interactions were performed in Biovia Discovery Studio Visualizer version 2016 (Yashoda Technical Campus, Satara, India) [27]. Visualization images were rendered by PyMOL 2.3 (Warren Lyford DeLano, 1 August 2006) [30].

### 2.5. In silico Pharmacokinetics (PK)/Pharmacodynamics (PD) Evaluation

Drug development requires many phases, starting with target identification and ending with ADMET prediction. Early detection of these characteristics is crucial to reduce the cost and time of the drug development process. To define the passage of this drug in the body, the measurement of the ADMET parameters of pharmacokinetics (Adsorption, Distribution, Metabolism, Excretion, and Toxicity) was performed. To this end, selected molecules based on their energy score were exploited to determine these in silico pharmacokinetic parameters using http://admetsar.com/ (Format SMILES accessed on 9 March 2020) to prevent the failure of these compounds in clinical trials and to increase their potential to reach the stage of drug candidates in the future.

### 2.6. Molecular Dynamics

Molecular dynamics simulations are more in-depth studies that explore the dynamism and conformational changes of bimolecular complexes. Molecular dynamics simulations tend to calculate the motions of atoms as a function of time by integrating the classical Newtonian equation of motion. The binding state of the ligand in the physiological environment has been anticipated by using simulations. In our study, compounds with higher affinity for the E6 active site and favorable pharmacokinetic characteristics F0679-0355, F33774-0275, and F3345-0326 (Figure 1) were selected and investigated by performing 50 nanosecond molecular dynamics (MD) simulations by using Desmond, a Schrödinger LLC software (New York, NY, USA) [31]. Protein–ligand complexes were preprocessed using Maestro’s Protein Preparation Wizard, which includes optimization and minimization of complexes. The System Builder tool was used to prepare all systems. TIP3P (Transferable Intermolecular Interaction Potential 3 Points) was chosen as the solvent model with an orthorhombic box. The OPLS_2005 force field was used in the simulation [32]. Counter ions were added to the models to make them neutral. Salt (NaCl) at 0.15 M was added to mimic physiological conditions. For the entire simulation, the NPT ensemble with a temperature of 300 K and a pressure of 1 atm was used. The models were relaxed before the simulation. Every 100 ps, the trajectories were saved for analysis, and the stability of the simulation was determined by measuring the root mean square deviation (RMSD) of the protein and ligand over time. The trajectories of the Desmond simulation were analyzed. The root mean square fluctuation (RMSF) and protein–ligand contacts were also calculated from the MD trajectory analysis.

## 3. Results and Discussion

### 3.1. Virtual Screening

In the present study, the main objective is the identification of potential new drugs as E6 inhibitors. A total of 6340 molecules that already have anticancer activity are processed by setting up drug-like filters to evaluate the similarity of these molecules with drugs, a high-throughput screening was reread through PyRx 1.0 software [33]. The chemical database was subjected to a simple filter to remove non-drug molecules to generate targeted databases for HPV16 E6. Accordingly, using Lipinski based filter, we have removed 962 non-suitable molecules chemicals allowing to narrow the database to 5378 drug-like compounds. 

The prepared chemical database was subjected to high-throughput screening to identify new candidate molecules potentially suitable for E6/E6AP/P53 inhibition, using AutoDoc vina involved in the PyRx tool which generated 9 distinct conformations for each ligand, ranked by binding affinity (kcal/mol). 

### 3.2. Molecular Docking

Autodoc vina and MGL tools were used to dock the 28 candidate molecules for HPV16 E6 treatment. This resulted in a final list of three compounds; F0679-0355, F3345-0326, and F3374-0275 showed the highest affinities: −10.4 Kcal/mol, −10 Kcal/mol, and −9.9 Kcal/mol, respectively. Total energy, hydrogen bonds (HBond) and other interactions of the three selected compounds are reported in Table 1. From the molecular docking results of the three candidate compounds, we found that the majority of these compounds share interactions with the same amino acids responsible for the formation of the E6/E6AP/p53 complex, and are involved in Van Der Waals interaction and pi-alkyl and pi-sigma interactions with residues Val31, Tyr32, Leu50, Cys51, Val53, Val62, Leu67, Tyr70, Ile73, Leu96, Cys97, Asp98, Leu99 and Leu100 of E6 contribute to the LxxLL binding pocket [34]. Whereas, Gln6, Glu7, Arg8, Arg10, Gln14, Glu18, Arg40, Glu41, Val42, Tyr43, Asp44, Phe45, Ala46, Phe47, Asp49, Leu50, Leu100, Cys106, Gln107, Lys108, Pro109, Leu110, Cys111, Pro112, Glu113, Gln114, and Lys115 of E6 [35] contribute to the p53 binding pocket which is consistent with recent studies (Table 1).

Molecular docking of the selected molecule F0679-0355 on HR E6 oncoprotein is reported in Figure 2 and showed that this ligand forms three strong hydrogen bonds with the amino acids Arg(131), Gln(107), and Tyr(32) of the HR E6 protein via 8 van der Waals bonds with the amino acids His(78), Ser(74), Ser(71), Arg(77), Tyr(70), Val(53), Val(31), and Phe(45). The docking study showed also the formation of 2 Halogen bonds between the ligand and the amino acids Cys(51) and Ala(61), and interacting via a π-Alkyl with the amino acids Ile(73) and Leu(50).

Results of *molecular docking* of the selected molecule F3345-0326 on HR E6 oncoprotein is reported in Figure 3 and clearly showed the formation of one strong hydrogen bond with the amino acids Ser(71) of the HR E6 protein and 9 van der Waals bonds with the amino acids Arg(129), His(78), Ser(74), Arg(131), Tyr(70), Leu(67),Val(53),Val(31),Gln(107), and 3 interacting via a π-Alkyl with the amino acids Val(62), Leu(50), and Ile(73). The docking study also showed the formation of 1 π-Sulfur bonds between the ligand and the amino acids Cys(51).

*Molecular docking* of the selected molecule F3374-0275 on HR E6 oncoprotein is reported in Figure 4 and showed the formation of two strong hydrogen bonds with the amino acids Cys(51) and Arg(102) of the HPV16 E6 protein and via 6 van der Waals bonds with the amino acids Gln(107), Leu(67), Tyr(32), Val(31), Phe(45), and Gly(130), and interacting via a π-Alkyl with the amino acids Val(53), Val(62), Leu(50), and Arg(131), and via π–sigma with Leu(100). The docking study showed also the formation of one carbon-hydrogen bond between the ligand and the amino acid Trp(132).

### 3.3. In Silico Pharmacokinetics (PK)/Pharmacodynamics (PD) Evaluation

The purpose of the *ADMET* preclinical study is to remove poorly performing molecules and focus on the best drug candidates. The present study investigated the suitability of the five proposed compounds for use as anti-cancer drugs based on properties, absorption, distribution, metabolism, excretion, and toxicity, which are critical elements in drug development.

These pharmacokinetic properties were obtained using the *admet SAR* approach evaluating blood-brain barrier (*BBB*) permeability, human intestinal absorption (*HIA*), Caco-2 cell permeability and the *AMES* assay that are the main elements used to determine drug capacities and results are reported in Table 2. Crossing the blood–brain barrier (*BBB*) is a crucial property widely used to determine the usefulness of the chemical as a drug, indicating whether drugs can cross the blood–brain barrier [36]. In this study, the three evaluated compounds have shown low *BBB* index and were, therefore, considered poorly distributed in the brain. 

*ADMET* analysis showed also that the three ligands can be absorbed by the human gut, are not metabolized by most *cytochrome P450* monooxygenases, and don’t exhibit any acute toxicity, mutagenic effect, or carcinogenic potential. 

### 3.4. Molecular Dynamics

In this study, extensive pharmacokinetic analysis identified three candidate compounds for HPV16 E6 inhibition with the most advantageous binding interactions and the best pharmacokinetic profiles. These three molecules have already shown anticancer activity the first F0679-0355 in sienna cancer F3345-0326 kinase inhibition F3374-0275 for leucime [37].

Desmond simulation trajectories were analyzed, and the root mean square deviation (RMSD) Figure 5, root mean square fluctuation (RMSF) Figure 6 and protein–ligand contacts were calculated from the MD trajectory analysis.

As reported in Figure 5, the evolution of RMSD values with time for the C-alpha atoms of three complexes *F0679-0355_4xr8*, *F3374-0275_4xr8*, and *F3345-0326_4xr8*, indicates that the three complexes reach stability at 10 ns. From then, changes in *RMSD* values remain within 1.0 Å for the target (*4xr8*) during the simulation period, which is quite acceptable. Ligands fit to protein *RMSD* values fluctuate within 1.5 Å till 50 ns after being stable. These indicate that the ligands remained stably bound to the binding site of the receptor during the simulation period. Of particular interest, F0679-0355_4xr8 complex showed better results comparatively.

Figure 6 manifests the *RMSF* value per residue of the target bound to the ligands. Residues showing larger peaks belong to the loop regions, as determined from the *MD* trajectories (Figure 7), or to the N- and C-terminal regions. The target bound to *F0679-0355* showed a comparatively better *RMSF*, as shown in Figure 6A. The low *RMSF* values of the binding site residues show the stability of the ligand binding to the protein.

The majority of the essential ligand–protein interactions determined by *MD* are hydrogen bonds and hydrophobic interactions, as illustrated in Figure 5. The stacked bar graphs have been normalized over the pathway: therefore, a value of 1.0 indicates that for 100% of the simulation time the particular interaction was conserved Figure 8. Values above 1.0 are likely to be achieved, as some protein residues could have multiple contacts of the same subtype with the ligand.

## 4. Conclusions

Using the in silico protein–ligand interaction approach, we identified three promising candidates with the highest binding energy scores that could be potential inhibitors of *HR E6* oncoproteins, likely without significant side effects. Furthermore, this study highlights the value of using the virtual screening approach as a time- and cost-saving strategy to identify chemicals with potential biological effects. However, further extensive in vitro and in vivo studies are needed to verify the antiviral activity of these three molecules.

In addition, to identify good molecules likely to bind to E6, but also to have medicinal properties, we have carried out an in silico study based on the identification of three molecules with anti-HPV activity.

## Figures and Tables

**Figure 1 vaccines-10-01452-f001:**
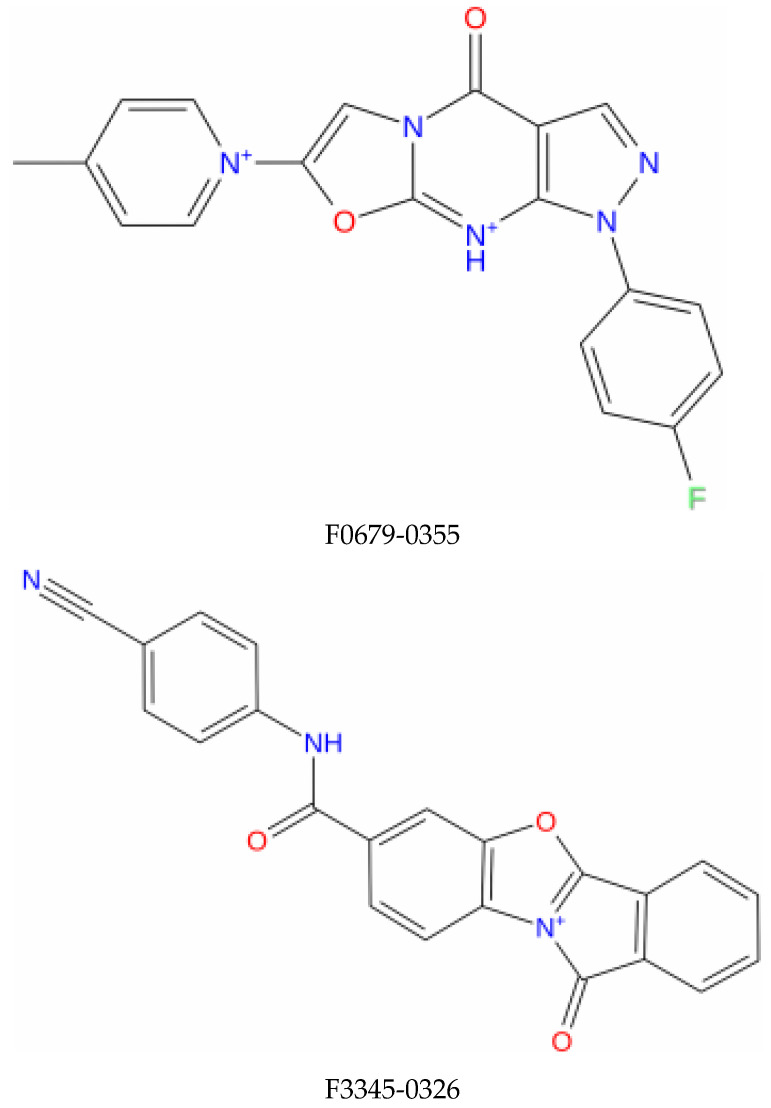
Structure of the top identified lead compounds.

**Figure 2 vaccines-10-01452-f002:**
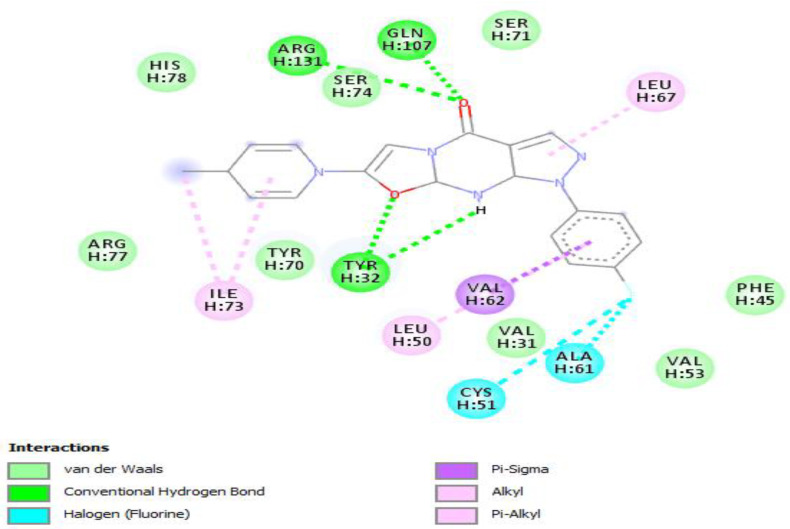
Positioning and interactions of the molecule F0679-0355 inside the active site of receptor HPV16E6 (PDB ID 4xr8).

**Figure 3 vaccines-10-01452-f003:**
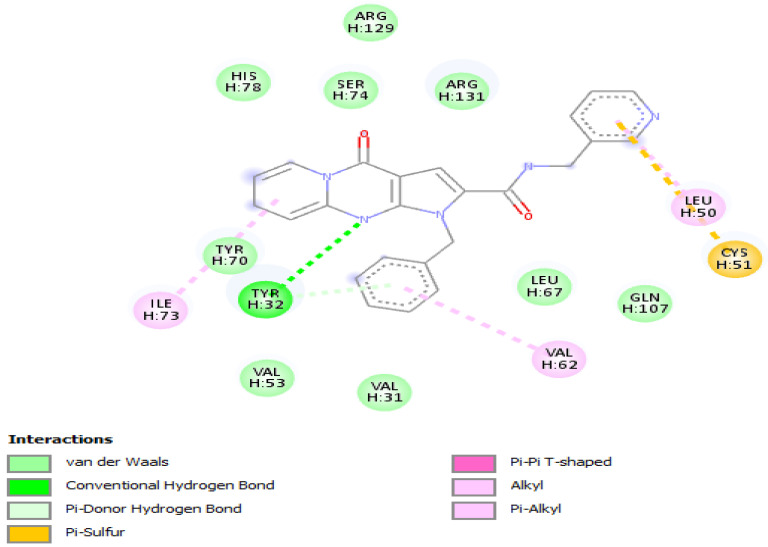
Positioning and interactions of the molecule F3345-0326 inside the active site of receptor HPV16E6 (PDB ID 4xr8).

**Figure 4 vaccines-10-01452-f004:**
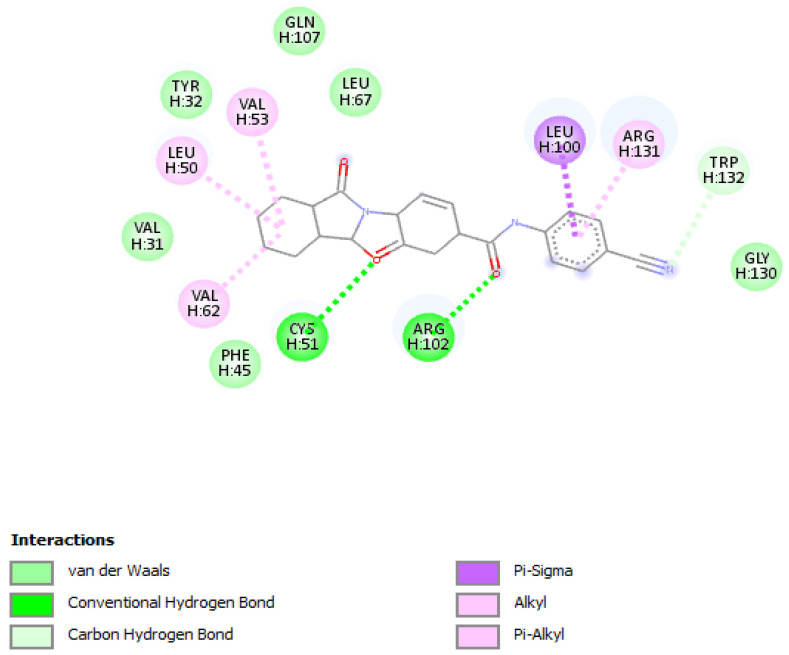
Positioning and interactions of the molecule F3374-0275 inside the active site of receptor HPV16E6 (PDB ID 4xr8).

**Figure 5 vaccines-10-01452-f005:**
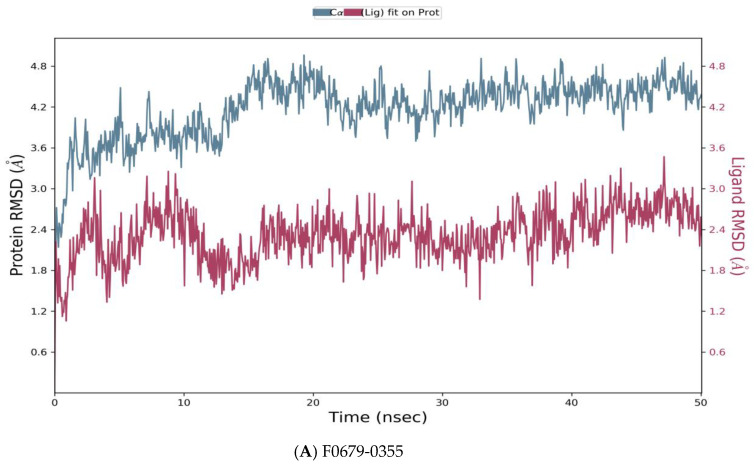
Root means square deviation (RMSD) of the C-alpha atoms of protein and the ligand with time. The left Y-axis shows the variation of protein RMSD through time. The right Y-axis shows the variation of ligand RMSD through time. (**A**) RMSD of F0679-0355_4xr8 complex. (**B**) RMSD of F3374-0275_4xr8 complex. (**C**) RMSD of F3345-0326_4xr8 complex.

**Figure 6 vaccines-10-01452-f006:**
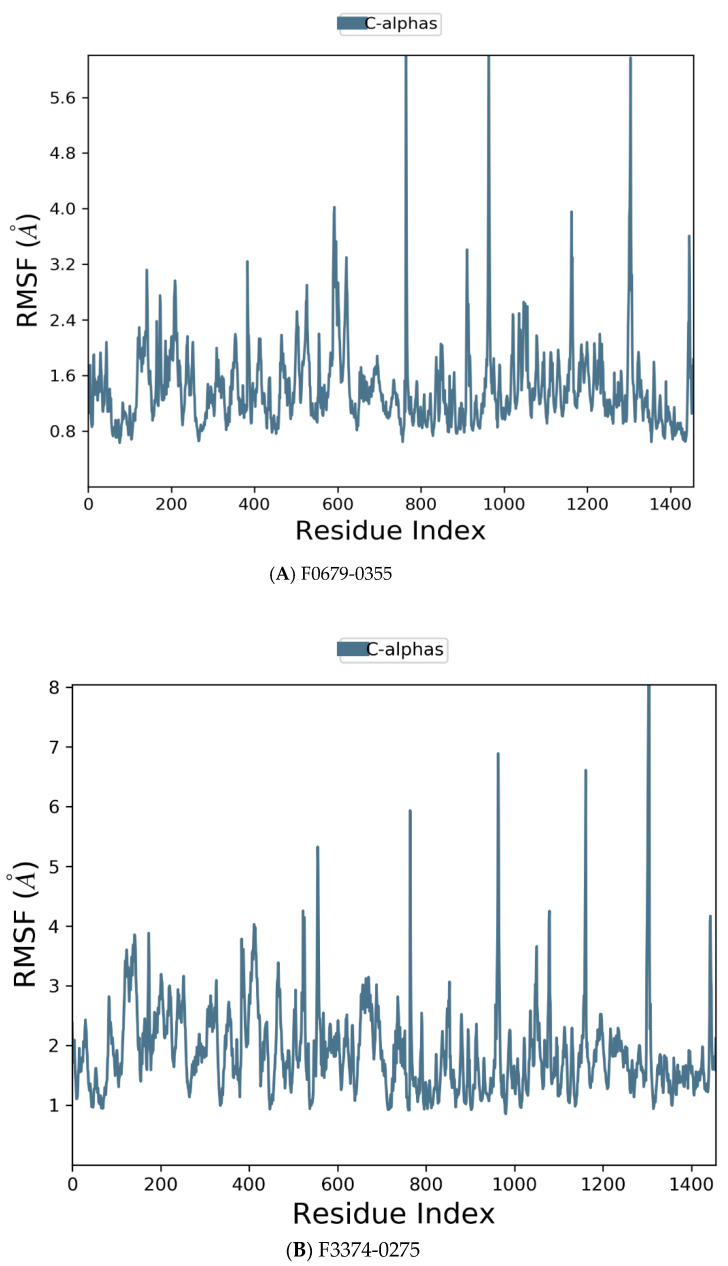
Residue wise Root Mean Square Fluctuation (*RMSF*) of protein with F0679-0355 (**A**), F3374-0275 (**B**), and F3345-0326 (**C**) ligands.

**Figure 7 vaccines-10-01452-f007:**
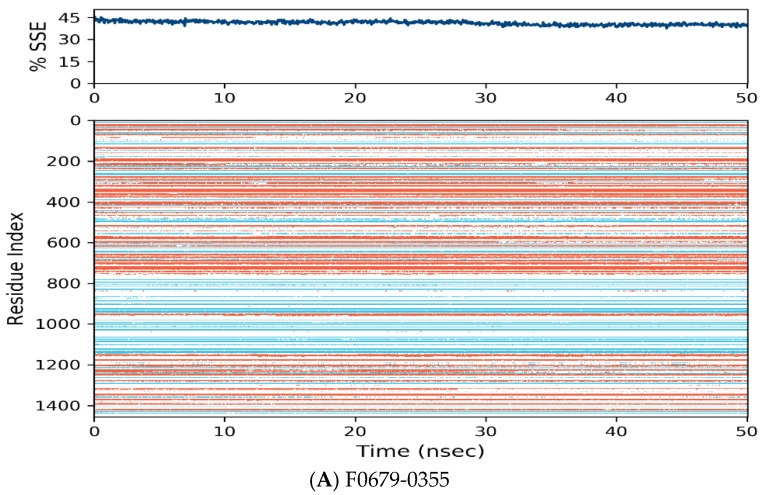
Distribution of protein secondary structure components by residue index across the protein structure. Red denotes alpha helices, and blue denotes beta-strands.

**Figure 8 vaccines-10-01452-f008:**
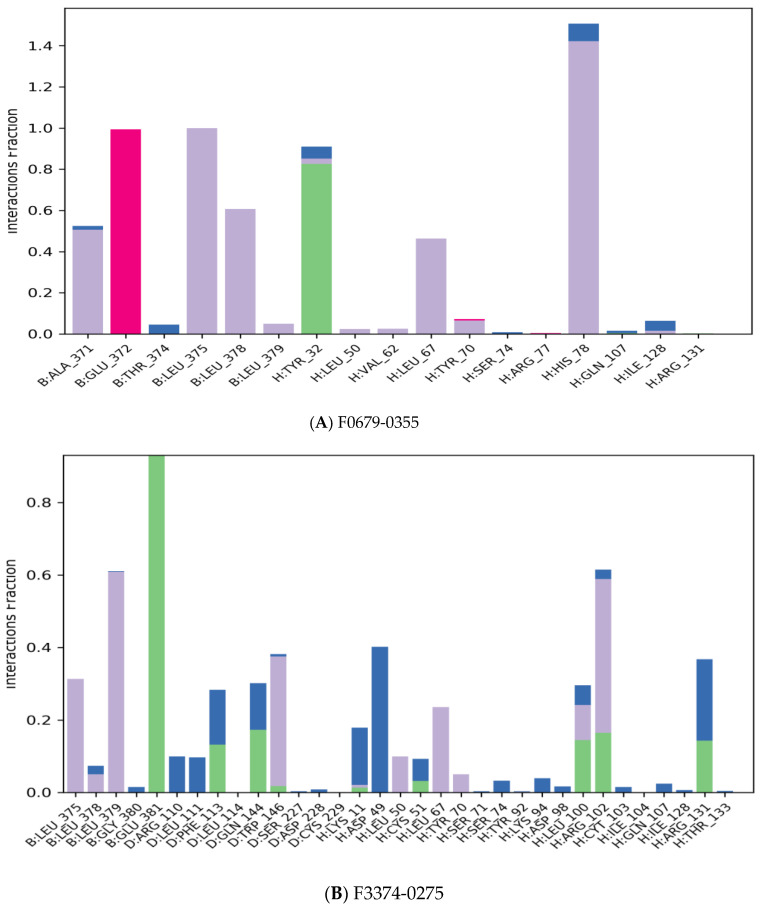
Protein–ligand contact histogram.

**Table 1 vaccines-10-01452-t001:** Molecular Docking result analysis of the top three compounds.

Compound Name	Molecular Formula	PubChem CID	Hydrogen Bonds	Other Interactions	Binding Affinity
**F3345-0326**	C_24_H_19_N_5_O_2_	4903840	Ser(71)	Val(53), Val(62), Leu(50), Cys(51), Phe(45), Val(31), Tyr(32), Ser(74), Ile(73), Arg(129), His(78), Tyr(70), Arg(131), Leu(67), Gln(107), Arg(102)	−10.4
**F3374-0275**	C_22_H_13_N_3_O_3_	17016408	Cys(51), Arg(102)	Gln(107), Leu(67), Tyr(32), Val(31), Val(31), Val(62), Phe(45), Val(53), Leu(50), Gln(130), Trp(132), Leu(100), Arg(131)	−10
**F0679-0355**	C_19_H_20_FN_5_O_2_	3155624	Arg(131), Gln(107), Tyr(32)	His(78), Ser(74), Ser(71), Leu(67), Arg(77), Ile (73), Tyr(70), Leu(50), Val(62), Val(31), Cys(51), Ala(61), Val(53), Phe(45)	−9.9

**Table 2 vaccines-10-01452-t002:** Analysis of the pharmacokinetic properties and toxicity of the three candidate molecules.

Compound Name	F0679-0355	F3345-0326	F3374-0275
**Absorption and Distribution**			
**Blood-Brain Barrier**	0.5177	0.6933	0.9385
**Human Gut Absorption**	0.9953	0.8780	0.9868
**Caco-2 Permeability**	0.6748	0.7412	0.6142
**Substratglycoprotéine P**	Yes	Yes	Yes
**Inhibitor of Glycoprotein P**	No	No	No
**Metabolism**			
**CYP450 2C9 Substrate**	No	No	No
**CYP450 2D6 Substrate**	No	No	No
**CYP450 3A4 Substrate**	Yes	Yes	Yes
**CYP450 1A2 Inhibitor**	No	No	No
**CYP450 2C9 Inhibitor**	No	Yes	No
**CYP450 2D6 Inhibitor**	No	No	No
**CYP450 2C19 Inhibitor**	No	Yes	No
**CYP3A4 Inhibitors**	No	No	No
**Excretion and Toxicity**			
**Hepatotoxicity**	No	No	No
**Carcinogens**	No	No	No
**AMES Mutagenicity**	No	No	No

## Data Availability

No new data were created or analyzed in this study. Data sharing is not applicable to this article.

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
