# Peer review of "In-Silico Molecular Modeling Studies to Identify Novel Potential Inhibitors of HPV E6 Protein"

_vaccines, 2022, doi:10.3390/vaccines10091452_

Round 1

Reviewer 1 Report

The manuscript vaccines-1825196 entitled “In-silico molecular modeling studies to identify novel Potential inhibitors of HPV E6 protein” by dr. Moujane, and colleagues reports on a computational approach aimed in identifying small molecules from 6346 chemicals, that already have anticancer activity, and available in the Life Chemicals database, with inhibitory activity on the E6-E6AP interaction. E6AP is ubiquitin ligase which complex with HPV E6 protein ultimately leading to tumor suppressor protein p53 degradation. A large variety of parameters were computationally evaluated. Three novel molecules were identified. Novel molecules represent a good starting point for the development of anti-HPV drugs.

The work is interesting, as a huge catalogue of molecules has been screened for their potential in targeting HPV16 oncogenic potential. However, the main limitation of the study is the lack of in vitro validation with cervical cancer cells. These compounds should be tested in vitro

A variety of modifications/improvements should be applied in this manuscript to making it suitable for publication (please see below)

My final recommendation is major revision

Major/general comments
The ms should be meticulously revised for the presence of several typo errors, while an English revision is also strongly recommended. Too many typo errors and English simplifications are present. For instance, among others, line 35 “it’s” should be “it is”, line 229 “Figure5”

Figures are difficult to read. Definition should be improved. The words on figures 4 and 7 should be enlarged for a better reading  

Study limitations should be included, in the discussion. For instance, the lack of in vitro validation

The discussion is quite poor and should be improved. If available, previous antitumor data on F0679-0355, F3345-0326 and F33774-0275, should be included in the discussion and detailed in the context of HPV oncogenic infection/HPV-driven tumors. The potential clinical application of these molecules should also be underlined

In silico, Papillomaviridae, as well as other Latinisms should be in italics style. Please revise the entire text, accordingly, including the title

Minor observations
Line 16 better “viral oncoprotein E5”
Line 33, better “500,000”
Line 41. As HPVs being sexually transmitted, these viruses are also involved in reproductive diseases https://doi.org/10.3390/vaccines8030473 and doi: 10.3390/v13122455 this information and supporting references should be included
Line 41 instead of “cancer” I suggest including the 5 HPV-driven human tumors, that are, vulvar, penile, anal, cervical and head and neck cancers PMID: 27127735

Reviewer 2 Report

In this manuscript Soumia et al. report the high throughput screening to identify candidate molecules potential suitable for drug treatment of HPV16 infections. Since the targeting of p53 by 16E6 is crucial for the oncogenic activity of HPV16, the authors screened for compounds that may interfere with the formation of the ternary complex E6-E6AP-p53. This finally resulted in the identification of 3 compounds. By molecular docking analysis using the data of the structure E6/E6AP/p53 from the RCSB protein data bank they modulate the interaction of these components with E6 for each of the three components. Since these putative contacts partially overlap with amino acids of HPV16E6 involved in p53 and E6AP targeting the authors conclude that the compounds may be valuable to act as a drug for treatment of HPV16 associated lesions.  

The identification of these three compounds is of interest for the field and may eventually result in the development in new drugs. However, the data are very preliminary. More important, more detailed information has to be given by the authors to allow the reader to follow the message of the manuscript. 

For instance, since the molecular and the structural details of the formation of the ternary complex E6-E6AP-p53 is crucial for the massage of this manuscript, the ternary complex including the residues of 16E6  involved in contact with p53  should be described in more detail in the introduction. Known characteristics of the amino acids putatively involved in the contact with each of the compound in the complex formation with E6AP and p53 should mentioned. This allows the reader to understand consequences of the drugs on the ternary E6/E6AP/p53 complex formation

There is no reference cited describing the structure of 16E6 and the ternary complex. In line 51, in addition to ref,. 9, the data of a more recent paper: Seiler et al., 2018, NATURE COMMUNICATIONS | DOI: 10.1038/s41467-018-06953-0 can be included as well as Martinez-Zapien et al., (2016, Nature Vol. 529, p. 541-545) discribing the crystal structure of the ternary complex E6-E6AP-p53. The results of these studies are crucial for this manuscript.

In line 167: it is mentioned that “the contacts of the compounds to E6 contributes to the p53 binding pocket which is consistent with recent studies”. These studies should be cited. 

The introduction is not well written. In addition to the missing information on the structural data on the ternary complex formation, as mentioned above,  further aspect have to improved. 

-It should be mentioned that HPV16 is also involved in head and neck cancer HNSC as well. 

-Line 37: the sentence is not complete

-Line 40: The statement “the interest has focused on 40 mucosal HPV genotypes” cannot be made, also cutaneous HP types have been investigated 

-Line 42: it is claimed that the genome consists of eight early  genes, yet in in brackets only 7 are numerated. 

-Line 44: “two late genes L1 and L2 are involved in the capsid formation” is slang and should be phrased correctly. 

The authors might include at least one experiment, which addresses the question whether any of the three compounds will interfere with the degradation of p53. 

The authors could treat HPV16 positive cell lines such as SiHa with their compounds and verify the level p53 by Western Blot. Alternatively, they may co-transfect cell lines such as HEK293 cells with epitope tagged vectors for 16E6 and p53 and analyze the level of p53 by Western Blot.  Expression vectors for 16E6 and p53 are available. 

In case the authors do not have the opportunity to work experimentally with cell cultures to perform such experiments, they can cooperate with a lab to do these well established experiments. In vivo experiments like these may very strengthen the manuscript and making it suitable for publication in a high ranked journal like viruses. 

 I

Round 2

Reviewer 2 Report

The authors have addressed the points. However, there remain several issues, which have to be corrected prior the  manuscript can be accepted for publication in Viruses.

1.     Lane 16: “oncoprotein virale” in French

2.     Lane 37: the sentence is not correct, something is missing

3.     The statement “HPV is  the etiological agent of HNSCC” is not correct. HNSCCs of the oral cavity and larynx are primarily associated with smoking and are referred to as HPV-negative HNSCC while tumors that arise in the oropharynx are linked to human papillomavirus (HPV). These are regarded as two different entities.  

4.     Lane 43/44: it should be: The two late genes L1 and L2 encode the structural proteins involved in capsid formation. 

5.     Lane 63: what does it mean p.300 in the brackets? Please correct.

6.     Lane 80: we identified 3 dry like components, resulting in the discovery of several chemical entities that offer – what does this sentence mean? Please clarify.

7.     Lane 238-240: what do the authors want to say with this sentence? The reference 37, which is cited here, describes the characterisation of three kinases. How does this reference refer to the sentence and this work? The compounds identified here are not mentioned in this manuscript. Please explain this and cite the correct reference. 

8. Lane 25: " In contrast, the identified molecules were selected." What do the authors want to say with this sentence in the abstract?

Author Response

Comments to reviewer 2

The authors would like to thank the editor and reviewers for their valuable time in evaluating our work. We have carefully processed all comments and taken your feedback into consideration. The color yellow represents changes and refinements made in the revised version. Please see below, in blue, our summarized responses to the comments.

Reviewer 2: 
Comments to the Author:

Dear Authors, The authors have addressed the points. However, there remain several issues, which have to be corrected prior the manuscript can be accepted for publication in Viruses.

  1. Lane 16: “oncoprotein virale” in French
  2. Lane 37: the sentence is not correct, something is missing
  3. The statement “HPV is the etiological agent of HNSCC” is not correct. HNSCCs of the oral cavity and larynx are primarily associated with smoking and are referred to as HPV-negative HNSCC while tumors that arise in the oropharynx are linked to human papillomavirus (HPV). These are regarded as two different entities.  
  4. Lane 43/44: it should be: The two late genes L1 and L2 encode the structural proteins involved in capsid formation. 
  5. Lane 63: what does it mean p.300 in the brackets? Please correct.
  6. Lane 80: we identified 3 dry like components, resulting in the discovery of several chemical entities that offer – what does this sentence mean? Please clarify.

  1. Lane 238-240: what do the authors want to say with this sentence? The reference 37, which is cited here, describes the characterisation of three kinases. How does this reference refer to the sentence and this work? The compounds identified here are not mentioned in this manuscript. Please explain this and cite the correct reference. 

  1. Lane 25: «In contrast, the identified molecules were selected." What do the authors want to say with this sentence in the abstract?

Authors’ Replies:

Thank you for your thorough review. It is our sincere hope that this report provides the necessary science to develop therapeutic strategies against HPV16. 

  1. Viral oncoprotein E6 has been identified as a key factor in cell immortalization and tumor progression in HPV-positive cells. E6
  2. And that high-risk (HR) HPVs are responsible for the occurrence and development of approximately 5% of all cancers and is associated with 30% of all pathogen-related cancers[1]

        [1]     M. Vonsky et al., « Carcinogenesis Associated with Human Papillomavirus Infection. Mechanisms and Potential for Immunotherapy  »,  Biochem. Mosc., vol. 84, no 7, p. 782‑799, juill. 2019, doi: 10.1134/S0006297919070095.

  1. Etiologic factors for developing HNSCC include the consumption of tobacco and alcohol, and high-risk human papillomavirus (HPV) infections[2]

         [2]    Y. Suh, I. Amelio, T. Guerrero Urbano, et M. Tavassoli, « Clinical update on cancer: molecular oncology of head and neck cancer », Cell Death Dis., vol. 5, no 1, p. e1018‑e1018, janv. 2014, doi: 10.1038/cddis.2013.548.

  1. And two late genes (L1 and L2) encode the structural proteins involved in capsid formation
  2. HR E6 oncoproteins can also interact with p300/CBP co-activators to control p53-dependent gene regulation [19].
  3. According to the database we have targeted, these molecules have already been used in the treatment of several cancers the three molecules that we have as potential inhibitors are already mentioned in the treatment of healthy cancer

  1. We have carried out the work on three drug-like compound successfully, which may have led to the discovery or proposed of other molecules that have the same basic structure of these three compounds studies which offer new scaffolds that could be used as the core of new familles of E6 Hpv16 inhibitors
  2. We proposed an in-silico approach for the discovery of small molecules with inhibitory activity on the E6-E6AP interaction. The first three compounds (F0679-0355, F33774-0275, and F3345-0326) were selected on the basis of virtual screening and prediction of the molecules' ADME properties and docking with E6 protein, these molecules were selected for further study by investigating their stability in the E6 complex and their inhibitory effect on the E6-E6AP interaction by molecular dynamics (MD) simulation. The identified molecules thus represent a good starting point for the development of anti-HPV drugs.